# Insight into the Volatile Profiles and Key Odorants of Rizhao Green Tea by Application of SBSE-GC-MS, OAVs and GC-O Analysis

**DOI:** 10.3390/foods14030458

**Published:** 2025-01-31

**Authors:** Mengqi Wang, Dapeng Song, Hongxu Yin, Fengxiang Fang, Yali Shi, Hui Wang, Jiyan Li, Kunpeng Wang, Yin Zhu, Haipeng Lv, Shibo Ding

**Affiliations:** 1Tea Research Institute, Rizhao Academy of Agricultural Sciences, Rizhao 276800, China; wmqrznky@163.com (M.W.); sdp20073882@163.com (D.S.); sdrzffx@163.com (F.F.); wh2009tea@126.com (H.W.); rzljy2663@163.com (J.L.); wkpahj888@163.com (K.W.); 2Rizhao Donggang District Agriculture and Rural Bureau, Rizhao 276800, China; sdjnyinhongxu@163.com; 3Tea Research Institute, Shandong Academy of Agricultural Sciences, Jinan 250100, China; shiyali97@163.com; 4Key Laboratory of Biology, Genetics and Breeding of Special Economic Animals and Plants, Ministry of Agriculture and Rural Affairs, Tea Research Institute, Chinese Academy of Agricultural Sciences, Hangzhou 310008, China; zhuy_scu@tricaas.com

**Keywords:** Rizhao green tea (RZT), SBSE-GC-MS, odor activity values, aroma quality, key odorants, aroma wheel

## Abstract

Rizhao green tea (RZT), a renowned green tea, is cultivated in China’s northernmost tea region. Its unique environment endows it with a strong chestnut- and seaweed-like aroma. This study sought to explore the volatile profiles of RZT and pinpoint its key odorants by employing stir bar sorptive extraction (SBSE) coupled with gas chromatography–mass spectrometry (GC-MS), determining the odor activity value (OAV), and performing gas chromatography–olfactometry (GC-O). A total of 112 volatiles were identified, and the major volatile compounds were esters (2035.25 μg/kg), alcohols (1799.02 μg/kg), alkanes (991.88 μg/kg), and ketones (691.96 μg/kg), comprising 74.91% of the total. A molecular aroma wheel was preliminarily established based on these key odorants. These insights might contribute to the scientific elucidation of the flavor chemical basis of RZT.

## 1. Introduction

Green tea is the most consumed and produced tea product in China, eith widespread appreciation from consumers around the globe for its delightful flavor. The quality of tea is intricately linked to factors such as the planting environment, climate, soil and geography [1]. Consequently, green teas from China’s northern regions exhibit distinctive flavor profiles that set them apart from their southern counterparts [2,3]. Rizhao green tea (RZT) is among the few highly esteemed green tea products cultivated in the northernmost tea region of China (35°04′ N–36°04′ N) (Figure 1). By 2023, Rizhao’s total tea planting area had grown to 20,000 hectares, producing over 19,800 tons of Rizhao green tea with an output value of CNY 3.8 billion. The unique growing conditions at this latitude, characterized by a humid maritime monsoon climate, low-temperature stress, and pronounced diurnal temperature fluctuations, result in the slower but more robust growth of RZT tea leaves and buds. Accordingly, these unique environments provide it with some special flavor characteristics different from those of the green tea grown in other tea regions in China, especially reflected in the RZT aroma quality: a rich chestnut- and seaweed-like aroma.

Aroma is a crucial parameter for assessing tea quality, primarily arising from the complex interaction of various volatile compounds in distinct proportions [4,5]. Research into the key odorants present in tea and the environmental factors influencing their synthesis can provide valuable insights for guiding tea cultivation and enhancing tea quality. RZT possesses special aroma quality that make it a valuable subject of study. However, the specific key odorants that shape its distinctive aroma and the biological processes behind their development remain unclear, which hinders the development of a comprehensive scientific understanding of how the aroma profile of RZT is linked to its cultivation environment.

It is widely recognized that different extraction methods can significantly influence the detection of tea volatiles in terms of both composition and quantity [6]. With high sensitivity and recovery, SBSE has been established as a practical and highly effective aroma enrichment technique that demonstrates high sensitivity and recovery. Presently, SBSE has gained significant traction in analytical applications, with widespread use in the analysis of tea, wine, and fruit [7,8,9]. In order to figure out the volatile profiles and clarify the key odorants of the green tea products produced in the Chinese northern tea regions, the volatile compounds of the representative RZT samples were analyzed in this study using SBSE in combination with GC-O and OAVs. The findings could enhance the understanding of green tea’s aroma quality across different cultivation conditions, as well as bridge the current research gap regarding the aroma quality of RZT, offering a theoretical framework for the quality assessment of RZT.

## 2. Materials and Methods

### 2.1. Tea Samples

Thirty-one Rizhao green tea samples (Table A1), commercially available and harvested from one bud and two leaves of fresh tea leaves (*Camellia sinensis* cv. Huangshan zhong), were sourced from the northern tea region, specifically Rizhao city in Shandong Province, China, and were gathered in extensive amounts in May 2021. Additionally, sensory evaluation of the above 31 RZT samples was performed according to the “Tea Sensory Evaluation Method” (GB/T 23776–2018) [10] and “Tea Sensory Evaluation Terminology” (GB/T 14487–2017) [11]. The sensory evaluation was conducted by 5 tea experts who are qualified as senior tea assessors, all of whom have more than 10 years of experience in tea sensory evaluation. Experts rated the tea samples for appearance, soup color, aroma, taste and the bottom of the leaves, and those with sensory scores <89 were removed. Ultimately, six tea samples that exhibited the most representative and quintessential flavor of RZT were finally selected as the subjects of this study. All tea samples were preserved in aluminum foil bags, shielded from light at −24 °C. Prior to analysis, the tea samples were ground for 40 s into powder at a speed of 24,000 rpm using a tube mill (CS025, IKA, Staufen, Germany). Additionally, a composite sample of the aforementioned six RZT samples was prepared for subsequent GC-O analysis.

### 2.2. Chemicals

Ethyl decanoate (≥98%) was purchased from Sigma-Aldrich (Shanghai, China). n-alkanes (C3–C7, C8–C40) were purchased from J&K Scientific (Beijing, China). Distilled water was purchased from Wahaha Group Company (Hangzhou, China). Aroma standards were purchased from J&K Scientific (Beijing, China) or Sigma-Aldrich (Shanghai, China), including hexanal (97%), 3-hexen-1-ol (98%), heptanal (98%), hexanoic acid, methyl ester (99%), 1-octen-3-ol (98%), 2-pentyl-furan (98%), octanal (98%), (E)-β-ocimene (≥95%), 3,5-octadien-2-one (95%), linalool (98%), nonanal (98%), phenethyl alcohol (98%), (Z)-3-nonen-1-ol (95%), methyl salicylate (98%), β-cyclocitral (95%), geraniol (98%), citral (99%), (E)-β-ionone (≥98%), indole (99%), (Z)-3-hexenyl hexanoate (99%), hexanoic acid, hexyl ester (99%), (Z)-Jasmone (99.0%), α-ionone (≥98%), geranylacetone (97%), phenylethyl alcohol (99.0%), dihydroactinidiolide (≥98%), and methyl jasmonate (95%).

### 2.3. SBSE Procedure

The volatiles were extracted using the SBSE technique, and the extraction conditions were based on those in the literature with slight modifications [12]. In detail, 600 mg of ground tea powder together with 500 mg of NaCl was weighed into a 20 mL sample bottle and brewed in 10 mL of boiling water. The PDMS twister (10 mm length, 1 mm thickness, 24 μL capacity, Gerstel, Germany) was immersed into the tea infusion. Then, the infusion was stirred and absorbed for 30 min at 80 °C and 1250 rpm on a multiposition hotplate stirrer (SP200-2 T; Miu Instruments Co., Ltd., Hangzhou, China). Finally, the twister was rinsed with purified water, wiped by nonwovens after extraction, and transferred to a thermal desorption tube for subsequent GC-MS analysis.

### 2.4. Thermal Desorption

The parameter settings for thermal desorption were based on those in the literature with small modifications [13,14]. For the GC–MS and GC–O analysis, the twister was inserted into the thermal desorption unit (TDU, Gerstel, Mülheim an der Ruhr, Germany) after extraction. The parameters used for the TDU were as follows: the desorption program was held at 30 °C for 1 min and raised to 240 °C (held for 5 min) at a rate of 100 °C/min, in splitless mode. The cooled injection system (Gerstel CIS-4 PTV injector) was maintained at −100 °C using liquid nitrogen (99.99%), and then raised to 280 °C (held for 3 min) at a rate of 12 °C/min sec from −100 °C (held for 1 min) after the desorption of the aroma volatile compounds.

### 2.5. GC-MS Analysis

The identification and quantification of volatiles were analyzed using an Agilent 7890B GC system coupled with an Agilent 5977B MSD mass spectrometer (Agilent, Santa Clara, CA, USA), equipped with an HP–5MS capillary column (30 m × 0.25 mm × 0.25 μm). The analytical method of GC-MS was based on previous studies [13,14]. The GC oven temperature program was follows: it started from 50 °C for 2 min, increased at 4 °C/min to 170 °C for 5 min, increased at 10 °C/min to 265 °C, and maintained for 5 min. The carrier gas was helium (>99.99%) with a constant flow rate of 1.6 mL/min, in solvent vent mode. Mass spectra were recorded in electron impact (EI) ionization mode at 70 eV. The ion source temperature was 220 °C. The interface temperature was set to 280 °C. The mass scan range was 30–600 AMU.

### 2.6. Identification and Qualification of the Volatile Compounds in RZT Samples

The volatile compounds were identified based on the NIST 2014 database library. The retention indices (RIs) of the compounds obtained from the experiment were compared with the theoretical values of the n-alkane (C3–C7, C8–C40) series under the same GC conditions. The quantification of the compounds was conducted based on their peak abundance in GC-MS analysis, with the relative content of the volatile compounds expressed in relation to that of ethyl decanoate, utilized as an external standard. For all the aroma volatile compounds extracted via SBSE, quantification was performed by comparing their peak areas to those of the external standard. In this study, the equation defining the external standard curve is as follows:*Y* = 12,766.062*X* − 111,889.342 (R^2^ = 0.996)
where R^2^ = 0.996 indicates a very high degree of correlation between *X* and *Y*. *Y* represents the abundance of volatiles’ chromatographic peaks, while *X* signifies the concentration of these volatiles, expressed in micrograms per kilogram (μg/kg).

### 2.7. OAVs Calculation

The odor activity value (OAV) is a standard metric used to assess the contributions of aroma compounds. The OAV is calculated as the ratio of the concentration of the compound in the tea infusion (C) to its odor threshold in water (OT). A compound is generally considered to have a vital contributing role to tea aroma quality, when the OAV is ≥1.

### 2.8. GC-O Analysis

The procedure for the GC-O analysis was based on that described in the literature [15]. The GC-O analysis was performed using GC-MS equipped with an ODP-2 olfactory detection port (Gerstel GmbH & Co. KG, Mülheim an der Ruhr, Germany). The volatiles extracted were split between the olfactory detection port and MS in a 1:1 ratio. The temperature of the GC-O injector was set at 230 °C and that of the transfer line was set at 250 °C. The carrier gas was high-purity nitrogen (99.99%). The temperature ramping procedure was the same as that described in Section 2.5.

GC-O analysis was conducted by a panel of experienced assessors, comprising five healthy, non-smoking individuals (three females and two males, all aged between 30 and 40). The GC-O analysis, in conjunction with the detection frequency method, was utilized to evaluate the aroma characteristics and odor intensities on a 4-point scale ranging from 1 to 4. This scale is defined as follows: 1 represents a weak intensity, 2 indicates a moderate intensity, 3 signifies a strong intensity, and 4 denotes an extremely strong intensity.

### 2.9. Statistical Analysis

All values were presented in the form of means. Figures were drawn using GraphPad Prism 9.0.0 software (GraphPad Software, San Diego, CA, USA), ChemDraw professional 16.0, and the OriginPro 2021 and online website (https://www.chiplot.online/, accessed on 25 August 2024).

## 3. Results and Discussion

### 3.1. Characterization of Aroma Compounds in RZT Identified by SBSE-GC-MS

As displayed in Table 1, a total of 112 volatile compounds from six representative and typical RZT samples were detected and identified by SBSE-GC-MS (Figure 2a). According to the differences in chemical structure, these volatiles can be further divided into 12 categories, including aldehydes, alkenes, esters, alcohols, oxygen heterocycles, nitrogen heterocycles, aromatics, phenols, alkanes, organic acids, and amines. Figure 2b illustrates the distribution of quantities and contents for a variety of volatile compounds in RZT. RZT had the greatest number of esters (22 types), followed by alkanes (21 types), alcohols (18 types), ketones (13 types), and so on. In addition, the primary volatile categories of RZT were esters (2035.25 μg/kg), alcohols (1799.02 μg/kg), alkanes (991.88 μg/kg) and ketones (691.96 μg/kg), which together accounted for 74.91% of the total volatile contents. In comparison, phenols (94.73 μg/kg) and amines (155.96 μg/kg) were present at the lowest concentration. As Figure 2c illustrates, twenty volatiles with high abundance (>100 μg/kg) were identified in RZT samples. Notably, (Z)-3-hexanoic acid hexenyl ester (433.71 μg/kg) and geraniol (406.46 μg/kg) were the most abundant. The former has an apple- and pear-like scent and was previously found in high abundance in XinYang MaoJian green tea [16]. It was discovered that (Z)-3-hexanoic acid hexenyl ester could increase the expression of the essential enzyme genes in the synthesis pathway, *CsADH1*, *CsADH3*, and *CsLOX3*, to strengthen the tea plant’s tolerance to the cold [17]. Geraniol, a colorless, oily terpene with a rose-like fragrance, is reported to be the most prevalent volatile in pan-fired green teas [12]. Linalool (292.88 μg/kg) was the third most abundant in RZT, sharing the same precursor (geranyl pyrophosphate, GPP) with geraniol [18]. It exhibits a distinct floral and sweet aroma, differing from that of geraniol, and is also abundant in various premium green tea products [5,19,20]. According to earlier research, cold stress can also cause a rise in the amounts of geraniol and linalool [21]. Although the exact process by which geraniol and linalool are inducted in tea is still unknown, it is hypothesized that the rise in these volatiles is connected to an alteration in the tea tree’s metabolic pathway at low temperatures.

In addition to the aforementioned, RZT contains a variety of other volatiles in higher concentrations, such as 1-butyl-2-isobutyl phthalate, α-linolenic acid, phytol, methyl jasmonate, (Z)-jasmone, indole, (Z)-3-hexenyl butanoate, hexadecanoic acid, jasmine lactone, (Z)-3-hexen-1-ol acetate, phenethyl alcohol, 2-ethyl-1-hexanol, (E)-β-ionone, (Z)-3-hexen-1-ol and others. Phytol, found in high abundance in Jingmai and Wuliang pu-erh green teas, has been reported to contribute to the aroma characteristics of green odor [22]. (E)-β-ionone, found in the majority of teas, possesses a scent reminiscent of violets. Both 3-hexen-1-ol and (Z)-3-hexenyl butanoate exhibit green flavor profiles; the former is crucial to the aroma of Longjing and Xinyang Maojian [16,23], while the latter is the predominant volatile and characteristic aroma compound of white tea [24]. Phenethyl alcohol has been detected in Laoshan green tea and imparts a rose-like scent [25]. 2-ethyl-1-hexanol contributes a citrus flavor and indole has floral aroma in a highly diluted form; both have been identified as key contributors to the chestnut-like aroma of Taiping Houkui green tea [26], and it is hypothesized that these two aforementioned volatiles may also have been factors contributing to the chestnut-like flavor of RZT samples in this research. α-linolenic acid is a representative unsaturated fatty acid in tea, and emits a waxy and fatty odor. Additionally, the unsaturated fatty acids serve as significant aroma precursors in tea, and their biosynthesis and degradation can be influenced by stress [18].

Figure 3 presents our hypothesis regarding the impact of low-temperature stress on fatty acid derivatives in RZT, based on the volatiles detected in this study. Research has shown that unsaturated fatty acids in tea can be oxidized by lipoxygenase (LOX) and subsequently decomposed into six-carbon aliphatic aromatic compounds. The metabolic pathways diverge based on the fate of hydrogen peroxide, an LOX oxidation product, into two branches: the LOX-AOS (lipoxygenase-allene oxygenase) and LOX-HPL (lipoxygenase-hydroperoxide lyases) pathways [27,28,29]. Unsaturated fatty acids can be converted into green leaf volatiles such as (Z)-3-hexen-1-ol and (Z)-3-hexen-1-ol acetate through oxidation, reduction, and esterification in the LOX-HPL pathway, and converted to jasmonic acid (JA) through oxidation and reduction in the LOX-AOS pathway [30]. Both pathways are significantly affected by environmental stresses. Low-temperature stress can enhance the accumulation of jasmonic acid in fresh tea leaves, and the concentration of its derivative, methyl jasmonate (MeJA), is also substantially elevated [31]. MeJA, known for its floral, fresh magnolia, and oily waxy scent, acts as a signaling factor that can induce changes in the levels of aroma precursors [32]. According to pertinent studies, the entire JA pathway in tea leaves can be rapidly activated by external MeJA, significantly affecting the biosynthesis of terpenoid backbones and leading to an increase in volatiles like linalool and geraniol [33]. External MeJA can also efficiently promote ROS (reactive oxygen species) scavenging and maintain cell membrane stability under cold stress [34]. Jasmine lactone, a cyclic volatile resulting from fat degradation [4], is characteristic of Oolong tea, and its accumulation in tea can be triggered by external stress such as low temperatures [35]. Therefore, it is hypothesized that the low-temperature stress caused by its high-latitude geographical location could be a major factor affecting the elevated levels of some unsaturated fatty acid derivative volatiles in RZT.

Although 1-butyl-2-isobutyl phthalate was abundant in RZT samples, there are no reports indicating its direct contribution to tea aroma. The same applies to hexadecanoic acid and alkanes; they are presumed to have negligible effects on the flavor of RZT samples due to their undetectable odor thresholds, and they cannot they be detected by GC-O analysis; however, it is hypothesized that these odorless volatiles may interact with other components such as amino acids and tea polyphenols, thereby influencing the flavor of tea.

### 3.2. Key Odorants in RZT Identified by OAV Analysis

As is well acknowledged, not all volatile compounds contribute to the aroma of tea. The quality and type of tea aroma are determined by the complex interaction of various volatile compounds at different concentrations, each with its own aroma contribution [15]. It is therefore insufficient to assess the contribution of aroma compounds to the aroma quality of tea just based on their content solely. The odor activity value (OAV) of a volatile reflects its influence on the overall aroma quality of tea; a compound with an OAV greater than 1 is typically considered a key odorant to the aroma and could help determine the overall aromatic quality [36]. Consequently, this study involved determining odor thresholds and calculating OAVs for 112 volatiles detected by GC-MS in order to screen out the key odorants that substantially influence the aroma quality of RZT. Table 1 presents the final computational and screening results. Out of the 45 volatiles analyzed, 26 were found to have OAVs greater than 1. These volatiles comprised seven aldehydes, two alkenes, six ketones, three esters, five alcohols, one oxygen heterocycle, one nitrogen heterocycle, and one aromatic. In terms of the aroma distribution of volatile compounds with an OAV > 1, the distinctive aroma profiles of RZT were characterized as floral (nine volatiles), fruity (five volatiles), and aldehydic (three volatiles), with cumulative OAVs of 2206.37, 121.74, and 250.85, respectively.

It is a well-established fact that a volatile compound can significantly influence the overall flavor profile of tea when its OAV exceeds 100 [37]. Among the identified volatiles, three volatiles had OAVs > 100, indicating their pivotal roles in the aroma of RZT: (E)-β-ionone (OAV = 1625.22), followed by phytol (OAV = 395.95), and decanal (OAV = 167.53). Notably, decanal is characterized by its fatty odor and is recognized as a key aroma compound in Chinese chestnuts [14,38]. Consequently, decanal is presumed to have a substantial effect on the chestnut-like aroma of RZT. Among other volatiles with OAVs > 10 in RZT, 3,5-octadien-2-one (OAV = 98.57) exhibits a fruity odor and is confirmed as a key odorant of Longjing tea [39]. Hexanal, heptanal, octanal, nonanal, 1-octene-3-ol, (Z)-jasmone, and jasmine lactone are primarily produced through the degradation of fatty acids [6], while β-Cyclocitral, (E)-β-ionone, α-ionone, and nerolidol are aromatic compounds derived from carotenoid cleavage [16]. Additionally, there exists a positive correlation between the quality of the chestnut-like aroma and the levels of (E)-β-ionone and nerolidol [40]. Other volatiles with OAVs > 1 include D-limonene, 1-octen-3-ol, 2-pentylfuran, hexanoic acid, methyl ester, methyl salicylate, indole, and (E)-β-ocimene, which have been identified as characteristic or aroma-active compounds in tea [7,15,41,42,43]. Additionally, previous research has indicated that D-limonene, 2-pentylfuran, and (E)-β-ocimene are the common aroma compounds of curled and pelleted RZT [44].

While research on the OAVs of tea volatiles has advanced, the limitations of OAV analysis have become increasingly apparent. For instance, certain volatiles exhibit a disproportionate relationship between their intensity and concentrations [7]. Moreover, the OAV alone is inadequate for accurately ranking the contributions of volatiles to a specific flavor profile. Some volatiles with lower OAVs might still be crucial to the overall flavor [45,46,47]. Consequently, to enhance OAV analysis, GC-O analysis which incorporates instrumental analysis and sensory determinations needs to be employed.

### 3.3. Key Odorants in RZT Identified by GC-O Analysis

GC-O, which uses the human olfactory system rather than an electronic nose, offers heightened sensitivity and is widely applied in contemporary research to detect a multitude of significant aroma compounds [48,49,50]. Compared to OAV analysis, GC-O is capable of evaluating the individual contributions of volatiles to the overall aroma, facilitating the identification of aroma-active compounds within complex mixtures [51].

By applying GC-O analysis, the sensory panel detected 37 key odorants from RZT samples, as indicated in Table 2. These odorants were categorized into seven groups (Figure 4a) based on their aroma types: floral and sweet (14 volatiles), fruity, citrus, and tropical (7 volatiles), green and minty (7 volatiles), aldehydic (4 volatiles), herbal (3 volatiles), cheesy (1 volatile), and earthy (1 volatile). Figure 4b shows that the floral and sweet category was the most prevalent in RZT, both in terms of the number of compounds and overall aroma intensities, with linalool exhibiting the highest aroma intensity (AI = 3.67). Followed by the green and minty aroma type, heptanal had the highest aroma intensity value (AI = 2.80). In the fruity, citrus, and tropical category, 2-pentylfuran had the highest aroma intensity (AI = 3.00). Among the aldehydic types, octanal and nonanal were found to have the most pronounced aroma intensity (AI = 3.00). Additionally, two unidentified chemicals (unknown 1 and unknown 2) with high aroma intensities that appeared as coconutty and floral and sweet were included in the samples. However, it is presumed that their concentrations in the tea were too low for structural identification. Therefore, the use of more sensitive instruments with lower detection limits, such as comprehensive two-dimensional gas chromatography coupled with time-of-flight mass spectrometry (GCxGC-TOFMS), is necessary for further systematic identification analysis.

### 3.4. Comparison of the Key Odorants in RZT Screened by OAV and GC-O

As anticipated, discrepancies exist between the OAV and GC-O results, as evidenced in Table 1 and Table 2. Some volatiles with lower OAVs exhibit stronger aroma intensities in GC-O, while certain volatile compounds with higher OAVs might not be detected at all. For example, phytol, with an OAV of approximately 400, was not identified in the GC-O analysis. Similarly, (E)-linalool oxide (furanoid), possessing an aroma intensity of 2.67, had an OAV less than 1. In actuality, the intricate process of volatile compound perception is affected by various factors, namely compound concentration, volatility, and the antagonism and synergy interactions between aroma compounds [52]. Studies on black tea, for instance, have demonstrated that methyl salicylate can exert diverse perceptual interaction effects on volatiles related to floral flavors (a masking effect, additive effect and synergistic effect) [53]. Calculating the OAV for a single chemical may not account for the complex reactions between aroma compounds. Therefore, it is hypothesized that these interaction effects could be responsible for the observed discrepancies in this study.

Despite minor discrepancies between the outcomes of GC-O and OAV analysis, the synergistic application of these two methodologies offers a more holistic portrayal of the aromatic profile of RZT. As depicted in Figure 5a, OAV and GC-O identified 25 and 37 key odorants, respectively, while the two groups mentioned above shared 17 common odorants. Therefore, the above 17 odorants, including linalool, geranial, (Z)-jasmone, α-ionone, (E)-β-ionone, indole, linalool, geranial indole, hexanoic acid methyl ester, 2-pentylfuran, 3,5-octadien-2-one, β-cyclocitral, methyl salicylate, heptanal, hexanal, octanal, nonanal, 1-octen-3-ol, and 2-heptanone, verified by OAV and GC-O methods, were considered the key odorants of RZT. The distinctive aroma quality of RZT arises from the complex interactions among these key odorants, which have thus been characterized as the key contributors to RZT’s aroma in this study. It is significant to note that prior research on Rizhao green tea’s aroma also identified linalool, hexanal, heptanal, octanal, and nonanal as the predominant compounds responsible for its chestnut-like aroma [54], aligning with our findings. The overall quantity of key odorants and volatiles of RZT found in the aforementioned study, however, slightly diverges from the findings of this investigation, this discrepancy being potentially attributable to differences in the methodologies employed for aroma extraction and detection.

In addition to the characteristic chestnut-like aroma of RZT, the volatile compounds responsible for its seaweed-like aroma are equally appealing. It has been proven that hexanal contributes to the formation of seafood flavors by providing its seaweed-like aroma [55] and is also identified as one of the principal compounds contributing to the aroma of Japanese matcha [56,57]. Additionally, 2-heptanone, found abundantly in various seafoods such as squid, seaweed, and crabs, is also a key aroma component of many seafood products [58,59,60]. Nevertheless, it exhibits a cream-like and cheese-like odor, which implies that 2-heptanone may contribute to the seaweed-like aroma of RZT through interactions with other components.

### 3.5. Establishment of Molecule Aroma Wheel of RZT

The aroma wheel served as a visual and intuitive tool for categorizing diverse types of flavor descriptions [61]. Currently, it is used in flavor research on beverages and condiments such as wine, coffee, and soy sauce [62,63,64]. The development of an aroma wheel can shed light on the chemical foundation of the distinctive aromatic qualities of RZT.

Building upon the sensory aroma wheel and the findings of previous studies as references [7,65], we refined the classification of key odorants according to their aroma types in order to more accurately define the aroma characteristics of RZT. The final RZT molecular aroma wheel is presented in Figure 6. The aroma quality of RZT was divided into seven groups using the molecular aroma wheel. Notably, the floral and sweet aroma type had the highest number of representative compounds, including linalool, geranial, (Z)-jasmone, α-ionone, (E)-β-ionone, and indole. Typical compounds of the fruity and citrus and tropical categories include hexanoic acid methyl ester, 2-pentylfuran, 3,5-octadien-2-one, and β-cyclocitral. Representative compounds of the green and minty category include methyl salicylate, heptanal, and hexanal. The aldehydic category includes octanal and nonanal, the earthy category contains 1-octen-3-ol, and the cheesy category comprises 2-heptanone. Through the integration of chemical composition and sensory descriptions, the molecular aroma wheel provides a comprehensive and consistent framework for evaluating RZT aromas, which can guide the creation of tea products with specific odor properties in future production.

However, as previously noted, the aroma quality of tea is not solely attributed to the presence of a single volatile; instead, it is the result of the comprehensive actions of multiple volatiles with significant aroma contributions. Moreover, these volatiles interact with one another through various mechanisms, including masking, additive, and synergistic effects. Therefore, the chestnut- and seaweed-like flavor characteristics of RZT do not suggest that the key odorants are confined to those that directly exhibit these flavors. Volatiles with floral, sweet, green, minty, aldehydic, and other odor types might also promote the perception of chestnut- and seaweed-like flavors in RZT. Consequently, future research on the flavor of RZT, including omission testing and other cutting-edge techniques, will be conducted to pinpoint the key odorants and ascertain their respective impacts.

While this study has yielded significant findings, several limitations must be acknowledged. Firstly, commercially available tea samples were used in this study, which may affect result reliability due to the slight differences in the production process. Additionally, the precursors and the biochemical pathways leading to the formation of these key odorants remain obscure, as do the regulatory mechanisms under the specific conditions of high latitude and low temperature. To address these limitations, and to conduct a more comprehensive investigation into RZT’s key odorants, future research should adopt a refined sampling strategy by processing the collected fresh tea leaves in a standardized technology. Moreover, sophisticated analysis technologies, including metabonomics and transcriptomics, will be essential to providing deeper insights into the underlying mechanisms.

## 4. Conclusions

Rizhao green tea (RZT) is renowned for its excellent aroma quality. This is the first comprehensive investigation to demonstrate the key odorants responsible for RZT’s unique flavor characteristic by conducting SBSE-GC-MS, OAVs and GC-O analysis. The findings indicate that RZT boasts a rich profile of volatile compounds, with alcohols, esters, alkanes, and ketones emerging as the predominant aromatic categories. Moreover, twenty-six and thirty-seven key odorants were identified by the OAVs and GC-O techniques, respectively, while seventeen key odorants, which were also confirmed through dual verification via OAVs and GC-O assessments, were revealed to be ultimately crucial in shaping RZT’s flavor profile. Furthermore, the RZT molecular aroma wheel was established based on these key odorants and their associated odor types. This research offers an in-depth analysis of RZT’s aroma characteristics, establishing a basis for the enhancement of processing techniques and the improvement of RZT’s aroma quality. Future research will examine how the amount of aroma precursors in tea leaves varies from winter to spring in order to shed light on how temperature and seasonal variations affect RZT’s aroma qualities at high latitudes. In addition, we will broaden the tea varieties and sample areas, and strive to provide a more thorough theoretical foundation for the investigation of the aroma characteristics of green tea in northern China.

## Figures and Tables

**Figure 1 foods-14-00458-f001:**
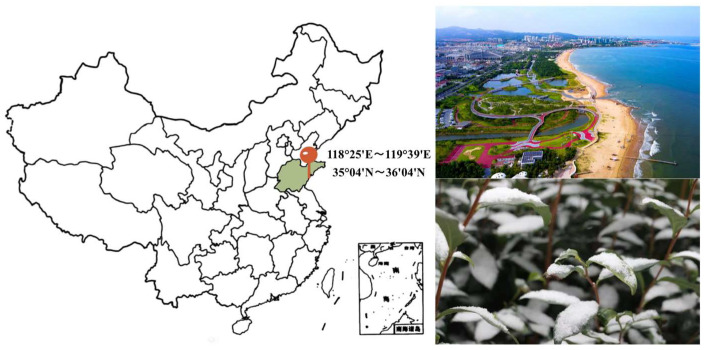
Geographical location and natural growth conditions of Rizhao green tea.

**Figure 2 foods-14-00458-f002:**
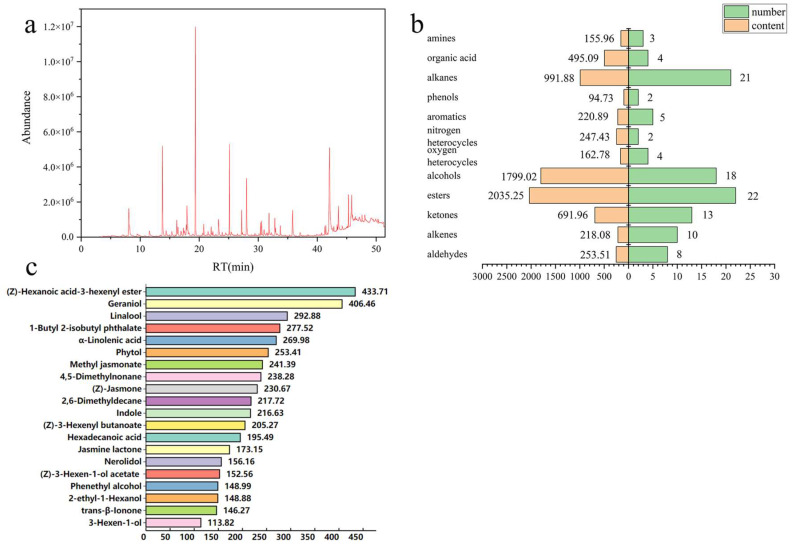
Result of volatile compounds in RZT identified by SBSE-GC-MS. (**a**) A representative total ion flow diagram of the RZT sample. (**b**) Volatile profiles of RZT samples. The left part of the figure shows the content of each volatile category, and the right part shows the number of volatiles in each category. (**c**) Details of volatile compounds with high contents (>100 μg/kg).

**Figure 3 foods-14-00458-f003:**
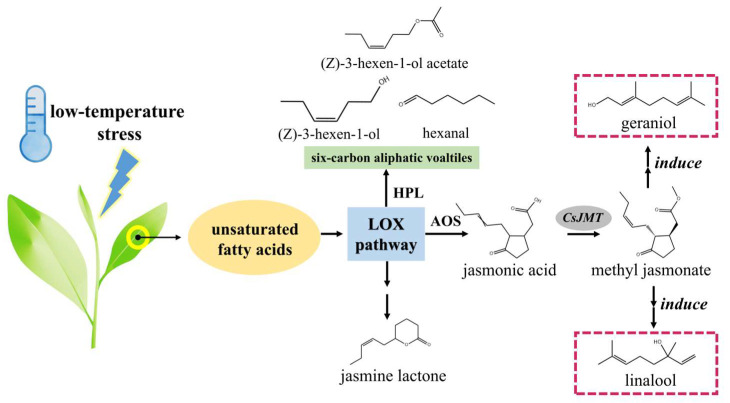
Effects of low-temperature stress on volatiles in fresh leaves of RZT (speculated). LOX: lipoxygenase; HPL: hydroperoxide lyases; AOS: allene oxygenase; JMT: JA carboxyl methyltransferase.

**Figure 4 foods-14-00458-f004:**
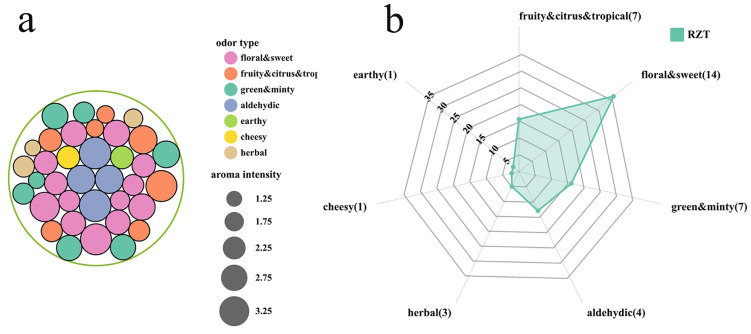
The results of GC-O analysis. (**a**) Circle packing plots of the odorants. (**b**) Radar map of the odorants. The *X*-axis represents the aroma type and number of corresponding compounds in that type, while the *Y*-axis represents the cumulative aroma intensity of different types of odorants.

**Figure 5 foods-14-00458-f005:**
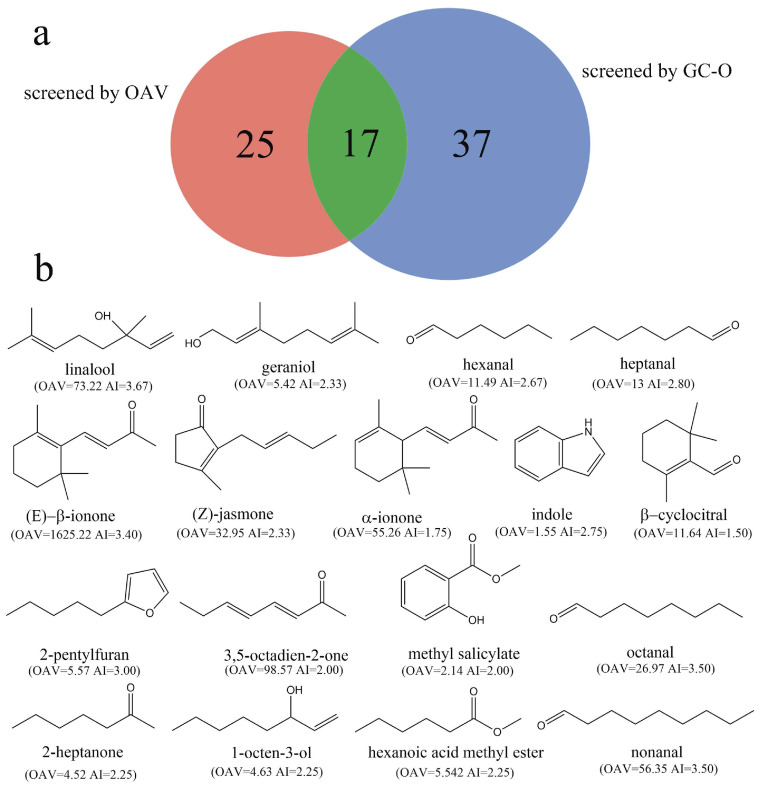
Key odorants based on OAV calculations and GC-O analysis in RZT. (**a**): a venn diagram of key odorants identified by OAV calculations and GC-O analysis. (**b**): chemical structure of 17 key odorants with dual verification of OAV and GC-O (the unknowns in GC-O cannot be categorized as key odorants because their names and chemical structures were not clear).

**Figure 6 foods-14-00458-f006:**
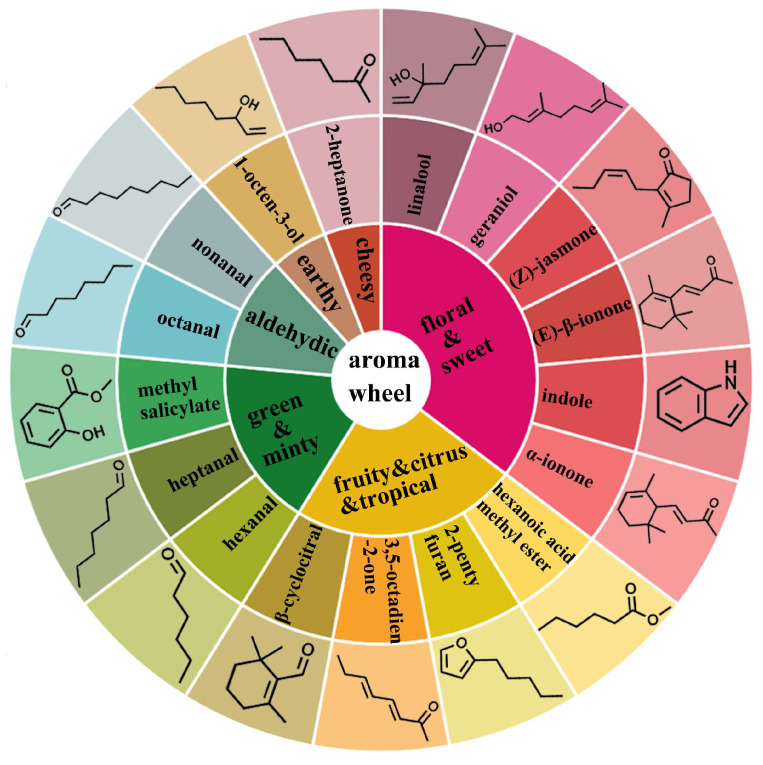
The molecular aroma wheel of RZT.

**Table 1 foods-14-00458-t001:** Volatile compounds of RZT identified by SBSE-GC-MS.

No	RT ^a^	Compounds	CAS	RI ^b^	Mean Content of RZT (μg/kg)	Content Range of RZT (μg/kg)	OT ^c^ (μg/kg)	OAV	Odor Type
Aldehydes									
1	7.74	Hexanal	66-25-1	800	51.70	47.62–55.80	4.5	11.49	Grass
2	10.79	Heptanal	111-71-7	903	39.00	34.72–44.76	3	13.00	Green
3	14.33	Octanal	124-13-0	1005	18.88	17.24–20.26	0.7	26.97	Aldehydic
4	16.14	2,6-Dimethyl-5-heptenal	106-72-9	1052	12.50	12.30–12.60	10	1.25	Melon
5	17.96	Nonanal	124-19-6	1106	56.35	49.11–61.05	1	56.35	Aldehydic
6	21.51	Decanal	112-31-2	1208	16.75	15.19–18.75	0.1	167.53	Aldehydic
7	22.18	β-Cyclocitral	432-25-7	1223	34.93	33.36–36.99	3	11.64	Tropical
8	23.83	Citral	5392-40-5	1276	23.40	21.65–24.42	62	<1	Citrus
SUM					253.51				
Alkenes									
1	8.81	2,4-Dimethylhept-1-ene	19549-87-2	843	20.12	18.71–23.94	NF	-	-
2	11.90	α-Pinene	80-56-8	940	16.58	14.94–19.14	180	<1	Herbal
3	13.87	α-Myrcene	123-35-3	993	22.65	20.78–24.60	44.5	<1	Spicy
4	14.10	(+)-4-Carene	29050-33-7	1002	11.39	9.94–13.04	NF	-	-
5	15.32	D-Limonene	138-86-3	1032	34.82	30.67–39.75	4	8.71	Fruity
6	15.94	(E)-β-Ocimene	3779-61-1	1051	22.03	20.48–23.30	18.7	1.18	Floral
7	17.55	4-Methyl-1-undecene	74630-39-0	1085	37.78	29.61–46.95	NF	-	-
8	18.37	(E)-4,8-Dimethylnona-1,3,7-triene	19945-61-0	1117	27.01	24.21–28.85	NF	-	-
9	26.40	α-Cubebene	17699-14-8	1353	13.52	12.99–14.64	NF	-	Herbal
10	27.72	β-cubebene	13744-15-5	1391	12.18	11.71–12.85	NF	-	Citrus
SUM					218.08				
Ketones									
1	7.81	4-methyl-3-Penten-2-one	141-79-7	802	50.31	46.90–54.60	NF	-	Vegetable
2	10.57	2-Heptanone	110-43-0	893	30.74	30.15–31.49	140	4.52	Cheesy
3	13.31	2,2-Dimethyl-3-heptanone	19078-97-8	967	22.53	19.36–29.29	NF	-	-
4	15.52	2,2,6-Trimethyl-cyclohexanone	2408-37-9	1041	28.45	27.17–29.90	NF	-	Spicy Cedar
5	17.02	3,5-Octadien-2-one	38284-27-4	1090	49.28	47.86–50.74	0.5	98.57	Fruity
6	22.93	2-Isopropyl-5-methyl-2-cyclohexen-1-one	5113-66-6	1251	9.98	9.74–10.40	NF	-	-
7	24.46	2-Undecanone	112-12-9	1294	23.43	21.09–24.67	7	3.35	Fruity
8	28.02	(Z)-Jasmone	488-10-8	1397	230.67	219.63–246.44	7	32.95	Floral
9	28.82	α-Ionone	127-41-3	1430	22.10	20.54–23.58	0.4	55.26	Woody, Floral
10	29.45	Geranylacetone	3796-70-1	1455	43.78	38.65–47.43	60	<1	Floral
11	30.52	4-(2,6,6-Trimethylcyclohexa-1,3-dienyl)but-3-en-2-one	1203-08-3	1485	17.29	16.80–17.94	NF	-	-
12	30.59	(E)-β-Ionone	79-77-6	1490	146.27	138.46–159.99	0.09	1625.22	Floral
13	30.74	2-Tridecanone	593-08-8	1501	17.13	16.11–17.80	10,000	<1	Waxy
SUM					691.96				
Esters									
1	11.61	Hexanoic acid, methyl ester	106-70-7	928	22.16	20.11–23.88	4	5.54	Fruity
2	14.40	3(Z)-Hexen-1-ol acetate,	3681-71-8	1007	152.56	144.25–163.41	NF	-	Green
3	20.76	(Z)-3-Hexenyl butanoate	16491-36-4	1189	205.27	186.02–219.25	20,000	<1	Green
4	21.05	(E)-Butanoic acid- 2-hexenyl ester,	53398-83-7	1196	19.15	17.63–19.98	NF	-	Green
5	21.47	Methyl salicylate	119-36-8	1194	85.68	77.20–93.51	40	2.14	Minty
6	22.32	(Z)-3-Hexenyl 2-methylbutanoate	53398-85-9	1234	76.61	69.87–82.52	NF	-	Green
7	26.48	Propanoic acid, 2-methyl-, 2-ethyl-3-hydroxyhexyl ester	74367-31-0	1373	37.78	29.61–46.95	NF	-	-
8	27.17	Propanoic acid, 2-methyl-, 3-hydroxy-2,2,4-trimethylpentyl ester	77-68-9	1380	68.00	54.05–88.04	NF	-	-
9	27.19	(Z)-Hexanoic acid-3-hexenyl ester,	31501-11-8	1383	433.71	425.88–438.43	NF	-	Green
10	27.33	Hexanoic acid, hexyl ester	6378-65-0	1386	19.49	17.90–20.15	6400	<1	Green
11	27.34	cis-3-Hexenyl cis-3-hexenoate	61444-38-0	1389	35.96	27.91–52.94	NF	-	Green
12	27.43	(E)-Hexanoic acid-2-hexenyl ester	53398-86-0	1391	21.68	20.33–22.65	NF	-	Waxy
13	29.23	β-Phenylethyl butyrate	103-52-6	1450	16.30	15.35–17.04	NF	-	Floral
14	30.99	Jasmine lactone	25524-95-2	1518	173.15	163.00–188.20	2	86.58	Creamy
15	32.22	Dihydroactinidiolide	17092-92-1	1542	61.41	58.56–62.81	NF	-	Musk Coumarin
16	33.04	Acetaminophen	1068-90-2	1602	21.46	19.72–22.63	NF	-	-
17	35.83	Methyl jasmonate	1211-29-6	1655	241.39	230.21–252.00	5700	<1	Floral
18	40.18	Isoamyl laurate	6309-51-9	1848	11.33	10.91–11.65	NF	-	Waxy
19	41.92	1-Butyl 2-isobutyl phthalate	17851-53-5	1933	277.52	240.18–320.20	NF	-	-
20	42.80	Hexadecanoic acid, 15-methyl-, methyl ester	6929-04-0	1984	29.11	19.01–43.86	NF	-	-
21	45.07	9,12-Octadecenoic acid, methyl ester	2462-85-3	2091	11.48	10.61–12.22	NF	-	-
22	45.15	Linolenic acid, methyl ester	301-00-8	2095	14.05	12.28–15.49	NF	-	-
SUM					2035.25				
Alcohols									
1	9.62	(Z-)3-Hexen-1-ol	928-96-1	857	111.82	103.84–123.67	NF	-	Green
2	10.06	1-Hexanol	111-27-3	872	17.14	15.73–20.01	200	<1	Green
3	13.63	1-Octen-3-ol	3391-86-4	982	64.78	61.84–67.61	14	4.63	Earthy
4	15.34	2-Ethyl-1-hexanol	104-76-7	1033	148.88	142.35–158.97	270,000	<1	Citrus
5	16.89	2-Furanmethanol, 5-ethenyltetrahydro-5-trimethyl-, cis-	104188-13-8	1066	69.08	64.60–73.70	NF	-	-
6	17.92	Linalool	78-70-6	1101	292.88	281.05–305.56	4	73.22	Floral
7	18.89	Phenethyl alcohol	60-12-8	1121	148.99	142.27–155.66	1200	<1	Floral
8	19.85	(Z)-3-Nonen-1-ol	10340-23-5	1159	12.02	11.69–12.50	NF	-	Waxy
9	20.44	1-Nonanol	143-08-8	1175	16.47	16.31–16.62	50	<1	Floral, Citrus
10	21.30	α-Terpineol	98-55-5	1191	21.69	20.63–23.67	280	<1	Terpenic
11	23.28	Geraniol	106-24-1	1258	406.46	402.09–415.01	75	5.42	Floral
12	30.15	1-Dodecanol	112-53-8	1477	15.23	13.75–16.69	73	<1	Waxy
13	31.02	(3S,3aR,3bR,4S,7R,7aR)-4-Isopropyl-3,7-dimethyloctahydro-1H	23445-02-5	1515	21.85	20.26–23.35	NF	-	Spicy
14	31.63	Bicyclo [3.1.0]hexan-2-ol, 5-[(1R)-1,5-dimethyl-4-hexen-1-yl]-2-methyl	58319-05-4	1540	11.84	11.10–12.57	NF	-	
15	32.82	Nerolidol	7212-44-4	1568	156.16	146.93–169.98	10	15.62	Floral
16	34.58	(-)-Torreyol	19435-97-3	1642	17.85	16.55–19.40	NF	-	Herbal
17	35.31	α-Cadinol	481-34-5	1653	12.47	12.31–12.73	NF	-	Herbal
18	45.31	Phytol	150-86-7	2109	253.41	220.29–292.77	0.64	395.95	Floral
SUM					1799.02				
Oxygen heterocycles									
1	20.60	(E)-linalool oxide (pyranoid)	39,028-58-5	1180	48.98	45.76–51.61	3000	<1	Woody
2	13.93	2-Pentyl-furan	3777-69-3	995	33.43	23.28–37.09	6	5.57	Fruity
3	13.94	(2R,5R)-2-Methyl-5-(prop-1-en-2-yl)-2-vinyltetrahydrofuran	54750-70-8	994	17.46	16.20–19.83	NF	-	-
4	17.46	(E)-Linalool oxide (furanoid)	34995-77-2	1091	62.91	58.10–70.62	190	<1	Woody, Floral
SUM					162.78				
Nitrogen heterocycles									
1	16.27	Tea pyrrole	2167-14-8	1053	30.80	29.06–32.38	65,000	<1	Roasted
2	25.35	Indole	120-72-9	1302	216.63	200.18–237.29	140	1.55	Floral
SUM					247.43				
Aromatics									
1	7.01	Toluene	108-88-3	771	82.69	75.05–94.57	NF	-	-
2	9.88	1,3-Dimethyl-benzene	108-38-3	873	65.94	59.29–82.55	41	1.61	Benzene-like
3	15.19	1-Methyl-3-(1-methylethyl)-benzene	535-77-3	1029	23.16	19.86–27.59	NF	-	-
4	31.34	Butylated Hydroxytoluene	128-37-0	1518	34.99	31.34–39.13	NF	-	Phenolic
5	28.49	1,6-Dimethylnaphthalene	575-43-9	1428	14.11	12.82–15.34	NF	-	-
SUM					220.89				
Phenols									
1	31.46	2,4-Bis(1,1-dimethylethyl)-phenol	96-76-4	1523	83.33	74.51–92.86	NF	-	-
2	36.42	Juniper camphor	473-04-1	1700	11.40	10.27–12.14	NF	-	-
SUM					94.73				
Alkanes									
1	8.22	2,4-Dimethyl-heptane	2213-23-2	822	90.53	85.76–98.94	NF	-	-
2	9.25	2,3-dimethyl-heptane	3074-71-3	856	15.57	14.41–19.23	NF	-	-
3	9.48	4-Methyl-octane	2216-34-4	864	59.15	54.37–67.28	NF	-	-
4	14.93	2,6-Dimethyl-nonane	17302-28-2	1020	31.60	23.80–42.38	NF	-	-
5	16.20	4,5-Dimethylnonane	17302-23-7	1057	238.28	220.13–254.99	NF	-	-
6	18.01	2,6-Dimethyldecane	13,150-81-7	1112	217.72	209.81–227.89	NF	-	-
7	18.71	3,7-Dimethyl-decane,	17312-54-8	1126	13.93	12.86–16.16	NF	-	-
8	21.24	4,7-Dimethyl-undecane,	17301-32-5	1208	33.16	25.34–41.45	NF	-	-
9	23.91	2,6,11-Trimethyl-dodecane	31295-56-4	1275	47.10	35.96–59.10	NF	-	-
10	24.64	2,3,5,8-Tetramethyl-decane,	192823-15-7	1318	21.34	17.30–25.02	NF	-	-
11	25.42	4,6-Dimethyl-dodecane,	61141-72-8	1325	35.34	28.16–46.36	NF	-	-
12	25.75	2,2,4,4,6,8,8-Heptamethylnonane	4390-04-9	1326	22.78	19.32–27.97	NF	-	-
13	26.08	1-Cyclohexylheptane	5617-41-4	1346	11.48	11.00–12.06	NF	-	-
14	26.30	2,6,10-Trimethyl-dodecane,	3891-98-3	1364	13.60	11.79–15.46	NF	-	-
15	26.87	3-Methyl-tridecane,	6418-41-3	1372	12.97	11.87–14.27	NF	-	-
16	27.78	6-Ethyl-2-methyldecane	62108-21-8	1390	24.05	18.99–28.36	NF	-	-
17	30.79	5-Propyl-tridecane,	55045-11-9	1502	20.56	18.50–22.65	NF	-	-
18	32.40	Nonylcyclohexane	2883-02-5	1558	15.46	14.25–16.47	NF	-	-
19	37.66	Phytane	638-36-8	1789	16.75	16.07–17.58	NF	-	-
20	40.48	2,6,10,14-Tetramethylheptadecane	18344-37-1	1872	17.79	15.65–21.63	NF	-	-
21	41.30	1,2-Epoxyoctadecane	7390-81-0	1900	32.72	23.14–43.23	NF	-	-
SUM					991.88				
Organic acid									
1	26.26	2-Methoxy-, methyl ester benzoic acid,	579-75-9	1347	11.93	11.68–12.28	NF	-	-
2	39.93	Pentadecylic acid	1002-84-2	1861	17.69	13.95–21.24	NF	-	Waxy
3	43.66	Hexadecanoic acid	57-10-3	1975	195.49	173.03–210.17	10,000	<1	Waxy
4	45.87	α-Linolenic acid	463-40-1	2119	269.98	241.30–291.66	NF	-	Fatty
SUM					495.09				
Amines									
1	24.97	N,N-dibutyl-formamide	761-65-9	1310	51.24	47.65–56.54	NF	-	-
2	42.25	5-Methyl-2-benzylhydrazide-3-isoxazolecarboxylic acid,	59-63-2	1945	14.81	14.00–15.23	NF	-	-
3	47.14	Hexadecanamide	629-54-9	2182	89.91	80.21–98.10	NF	-	-
SUM					155.96				

^a^ Retention time. ^b^ The retention index (RI) from the published literature and an online library (https://webbook.nist.gov/chemistry/cas-ser.html, accessed on accessed on 5 July 2024); ^c^ odor threshold, all the odor thresholds were obtained from the literature (Fenaroli’s handbook of flavor ingredients). -, no odor description information was found in the literature. NF, not found in the literature.

**Table 2 foods-14-00458-t002:** Key odorants in RZT identified by GC-O analysis.

No.	TR (min) ^d^	Aroma Compounds	Odor Descriptors	AI ^e^	IM ^f^
1	7.72–8.01	Hexanal	Grass, leafy	2.67	MS ^g^, RI ^h^, A ^i^, O ^j^
2	9.63–9.77	3-Hexen-1-ol	Green, grassy	1.33	MS, RI, A, O
3	10.45–10.57	2-Heptanone	Cheese, coconut	2.25	MS, RI, A
4	10.80–11.23	Heptanal	Fresh, green	2.80	MS, RI, A, O
5	11.50–11.97	Hexanoic acid, methyl ester	Fruity, sweet	2.25	MS, RI, A, O
6	13.45–13.78	1-Octen-3-ol	Green, metallic	2.25	MS, RI, A, O
7	13.75–14.07	2-Pentyl-furan	Fruity, green	3.00	MS, RI, A, O
8	14.32–14.57	Octanal	Aldhydic, waxy	3.50	MS, RI, A, O
9	15.04–15.15	2-Ethyl-1-hexanol	Sweet, herbal, green	1.50	MS, RI, A, O
10	16.19–16.29	(E)-β-Ocimene	Sweet, floral	2.00	MS, RI, A, O
11	16.95–17.23	3,5-Octadien-2-one	Fruity, green	2.00	MS, RI, A, O
12	17.46–17.83	(E)-Linalool oxide (furanoid)	Floral, coffee, baked	2.67	MS, RI, A
13	17.53–18.14	Linalool	Floral, sweet	3.67	MS, RI, A, O
14	17.89–18.14	Nonanal	Waxy aldehydic	3.50	MS, RI, A, O
15	18.89–19.12	Phenethyl alcohol	Sweet, floral	2.50	MS, RI, A, O
16	19.85–20.13	(Z)-3-Nonen-1-ol	Mushroom, green, spicy	3.00	MS, RI, A, O
17	20.49–20.72	(E)-Linalool oxide (pyranoid)	Fresh, woody	3.17	MS, RI, A
18	21.41–21.45	Methyl salicylate	Wintergreen, minty	2.00	MS, RI, A, O
19	22.12–22.46	β-Cyclocitral	Fresh, clean, citrus	1.50	MS, RI, A, O
20	22.38–22.99	(Z)-3-Hexenyl 2-methylbutanoate	Green, apple, sweet	2.00	MS, RI, A
21	23.28–23.49	Geraniol	floral, sweet	2.33	MS, RI, A, O
22	23.93–24.09	Citral	Sweet, citrus, fruity	2.00	MS, RI, A, O
23	25.34–25.56	Indole	Floral, sweet, plastic	2.75	MS, RI, A, O
24	26.50–26.66	α-Cubebene	herbal	2.00	MS, RI, A,
25	26.79–27.05	(Z)-3-Hexenyl hexanoate	Citrus, fruity,	2.60	MS, RI, A, O
26	27.44–27.76	Hexanoic acid, hexyl ester	Fruity, green, sweet	2.60	MS, RI, A, O
27	28.04–28.38	(Z)-Jasmone	Floral, sour, sweet	2.33	MS, RI, A, O
28	28.85–29.04	α-Ionone	Floral, violet	1.75	MS, RI, A, O
29	29.25–29.75	Geranylacetone	Magnolia, sweet	2.00	MS, RI, A, O
30	29.98–30.29	1-Dodecanol	Fatty, waxy, sweet	3.00	MS, RI, A
31	30.51–30.79	(E)-β-Ionone	Floral, orris	3.40	MS, RI, A, O
32	31.06–31.53	Unknown 1	coconut	3.40	-
33	31.87–32.25	Dihydroactinidiolide	Floral, sweet, fresh	2.50	MS, RI, A, O
34	34.46–34.70	(-)-Torreyol	Fresh, sweet, herbal	1.25	MS, RI, A
35	35.33–35.61	α-Cadinol	Herbal. green, mild sweet	1.67	MS, RI, A
36	35.83–36.11	Methyl jasmonate	Floral, sweet, milk	2.25	MS, RI, A, O
37	37.11–37.43	Unknown 2	Floral	2.75	-

^d^: time range. ^e^: aroma intensity. ^f^: identification method (IM). ^g^: mass spectrum (MS), compared with the Nist library. ^h^: retention index (RI), compared with the RI in the published literature and an online library (https://webbook.nist.gov/chemistry/cas-ser.html, accessed on 5 July 2024). ^i^: aroma descriptors (A). ^j^: (O) odor of the authentic standard.

## Data Availability

The original contributions presented in this study are included in the article. Further inquiries can be directed to the corresponding authors.

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
