# Peer review of "Insight into the Volatile Profiles and Key Odorants of Rizhao Green Tea by Application of SBSE-GC-MS, OAVs and GC-O Analysis"

_foods, 2025, doi:10.3390/foods14030458_

Round 1

Reviewer 1 Report

Comments and Suggestions for Authors

Dear authors, the revised manuscript is interesting. This study sought to explore the volatile profiles of RZT and pinpoint its key odorants by employing stir bar sorptive extraction (SBSE) coupled with gas chromatography-mass spectrometry (GC-MS), odor activity value (OAV), and gas chromatography-olfactometry (GC-O). In this first review the following recommendations are made:

Line 21: insert text space… 2035.25 μg/kg

Line 21: insert text space… 1799.02 μg/kg

Line 22: insert text space… 991.88 μg/kg

Line 22: insert text space… 691.96 μg/kg

Line 32: could include information on total production or per capita consumption of a specific region

Line 38: rewrite… (Figure 1).

Line 60: use [79] instead of [7-9]

Line 69: Thirty-one RZT samples?

Line 70: use italic text format for scientific names

Line 76: equipment information (model, brand, country)

Line 97: 1 mm

Line 99: 80°C, text format like in line 75

Line 107: 30°C

Line 107: 240°C

Line 108: 100°C

Line 109: -100°C

Line 109: 280°C

Line 110: 12°C

Line 105-111: Is a published protocol followed to perform this procedure? If so, add the reference

Line 117: 50°C

Line 118: 4°C

Line 118: 170°C

Line 118: 265°C

Line 121: 220°C

Line 121: 280°C

Line 114-122: Is a published protocol followed to perform this procedure? If so, add the reference

Line 150: 230°C

Line 150: 250°C

Line 161: use presented instead of displayed

Line 161: use means instead of averages

Line 174: insert text space… 2035.25 μg/kg

Line 174: insert text space… 1799.02 μg/kg

Line 174: insert text space… 991.88 μg/kg

Line 175: 691.96 μg/kg

Line 176: 94.73 μg/kg

Line 176: 155.96 μg/kg

Line 178: 433.71 μg/kg

Line 179: 406.46 μg/kg

brackets.

Line 182: 292.88 μg/kg

Line 188-190: align ident text

Line 190: > 100 μg/kg

Line 194: In the last column of the table, start the first letter of each word with a capital letter, and insert text space where required.

Line 317: insert a dot at the end of table title

Line 324,325: align ident text

Line 355: align ident text

Line 370: [5860].

Line 356-357: align ident text

Line 385: rewrite… Figure 6. The….

Line 447: Journal names in references should be abbreviated (correct through this section)

Line 447: use italic text format for scientific names (correct through this section)

Line 448: Only the volume of the journal should appear in italic text format; remove the issue number (correct through this section)

Line 449: References appear in which the authors' information is incomplete (correct through this section)

Note: It is necessary to carefully review and correct the format of the references section

What is the main question addressed by the research?

This research addresses the main question: What are the key volatile profiles and odorants of Rizhao green tea (RZT), and how do they contribute to its sensory characteristics? This question seeks to unravel the chemical and sensory profile of RZT tea using advanced techniques (SBSE-GC-MS, OAV, and GC-O) to understand the compounds that define its organoleptic properties.

Do you consider the topic original or relevant to the field? Does it address a specific gap in the field? Please also explain why this is/ is not the case.

The topic is original and relevant because, although there are studies on volatile profiles in green teas from different regions, RZT tea has unique characteristics due to its geographical location and production methods. This work fills a gap in the knowledge of its volatile compounds and their relationships with sensory attributes. The results can influence production, quality improvement, and product differentiation in competitive markets. The combined use of SBSE-GC-MS, OAV, and GC-O is a robust methodology not explored explicitly in RZT, adding value to the study.

What does it add to the subject area compared with other published material?

Beyond listing volatile compounds, the focus on OAV and GC-O allows us to identify which directly impacts the organoleptic characteristics of RZT tea. Combining SBSE with GC-MS and GC-O provides a deeper and more precise analysis than conventional techniques. By focusing on RZT tea, the work not only contributes to the general knowledge of tea but also valorizes a local product with potential in global markets.

What specific improvements should the authors consider regarding the methodology?

In this first review, the authors were recommended to indicate the origin of some sections of the methodology or whether they are their methods.

Are the conclusions consistent with the evidence and arguments presented and do they address the main question posed? Please also explain why this is/is not the case.

In this first review, the authors were told that the results are presented and discussed broadly and are consistent with the work's conclusions.

Are the references appropriate?

In this first review, the authors were recommended to review and correct the format of the references section carefully.

Author Response

Comments 1: Line 21: insert text space… 2035.25 μg/kg; Line 21: insert text space… 1799.02 μg/kg; Line 22: insert text space… 991.88 μg/kg; Line 22: insert text space… 691.96 μg/kg;

Response 1: Thanks for your careful check, we have modified the format of data and units accordingly.

Comments 2: Line 32: could include information on total production or per capita consumption of a specific region

Response 2: thanks for your kind suggestion! Relevant data on the output value of Rizhao green tea have been added to the introduction, details can be seen in line 40-42.

The content added is: By 2023, Rizhao's total tea planting area had grown to 20,000 hectares, producing over 19,800 tons of Rizhao green tea with an output value of 3.8 billion yuan. Data from Rizhao Tea Association (2023 Rizhao Tea Industry Annual Inventory)

Comments 3: Line 38: rewrite… (Figure 1).

Response 3: Have been modified

Comments 4: Line 60: use [7–9] instead of [7-9]

Response 4: Have been modified

Comments 5: Line 69: Thirty-one RZT samples?

Response 5: Thanks for the reviewer's suggestion. In fact, a total of 31 samples were collected at the beginning, the details of which can be seen in Table A1 of the article. After that, we organized 5 tea experts to conduct sensory evaluation on these 31 tea samples, and finally 6 RZT samples with representative flavor were screened. In order to ensure the rigor of the experiment, we have added the sensory evaluation reference criterias and the process to the revised manuscript, which you can see on line 80-85.

The content added is: Sensory evaluation of the above 31 RZT samples was performed according to the ”Tea Sensory Evaluation Method” (GB/T 23776–2018) and “Tea Sensory Evaluation Terminology” (GB/T 14487–2017). The sensory evaluation was conducted by 5 tea experts who are qualified as senior tea assessors, all of whom have more than 10 years of experience in tea sensory evaluation. Experts rated the tea samples for appearance, soup color, aroma, taste and the bottom of the leaves, and those with sensory scores < 89 were removed. Ultimately, six tea samples that exhibited the most representative and quintessential flavor of RZT were finally selected as the subjects of this study.

Comments 6: Line 70: use italic text format for scientific names

Response 6: Thanks to your suggestion, we have changed the format of the scientific name.

Comments 7: Line 76: equipment information (model, brand, country)

Response 7: The grinding machine was purchased from IKA Company in Germany, model CS025. The changes can be seen on line 88.

Comments 8: Line 97: 1 mm

Response 8: Thanks for your careful check, we have modified the format of data and units accordingly.

Comments 9: Line 99: 80°C, text format like in line 75

Response 9: Thanks for your careful check, we have modified the format of data and units accordingly.

Comments 10: Line 107: 30°C; Line 107: 240°C; Line 108: 100°C; Line 109: -100°C; Line 109: 280°C; Line 110: 12°C;

Response 10: Thanks for your careful check, we have modified the format of data and units accordingly.

Comments 11: Line 105-111: Is a published protocol followed to perform this procedure? If so, add the reference

Response 11: We appreciate the reviewers' recommendation. There are references that correspond to the thermal desorption procedure. In order to ensure the persuasions of the research, the author has added appropriate references into the revised manuscript. Details of the references are as follows:

[11] Zhu, Y., Lin, Z., Shao, C., Zhang, Y., Lv, H., & Zhang, Z. Aromatic profiles and enantiomeric distributions of chiral odorants in baked green teas with different picking tenderness. Food Chem,2022,15, 388.DOI:10.1016/j.foodchem.2022.132969.

[12] Wang, M. Q., Ma, W. J., Shi, J., Zhu, Y., Lin, Z., & Lv, H. P. Characterization of the key aroma compounds in Longjing tea using stir bar sorptive extraction (SBSE) combined with gas chromatography-mass spectrometry (GC–MS), gas chromatography-olfactometry (GC-O), odor activity value (OAV), and aroma recombination. Food Res Int,2020,130, 108908.https://doi.org/10.1016/j.foodres.2019.108908.

Comments 12: Line 117: 50°C; Line 118: 4°C; Line 118: 170°C; Line 118: 265°C; Line 121: 220°C; Line 121: 280°C

Response 12: Thanks for your careful check, we have modified the format of data and units accordingly.

Comments 13: Line 114-122: Is a published protocol followed to perform this procedure? If so, add the reference

Response 13: We appreciate the reviewers' recommendation. There were references[11,12] that correspond to the GC-MS procedure. The references were consistent with those for the TDU procedure, and had added into the revised manuscript.

[11] Zhu, Y., Lin, Z., Shao, C., Zhang, Y., Lv, H., & Zhang, Z. Aromatic profiles and enantiomeric distributions of chiral odorants in baked green teas with different picking tenderness. Food Chem,2022,15, 388.DOI:10.1016/j.foodchem.2022.132969.

[12] Wang, M. Q., Ma, W. J., Shi, J., Zhu, Y., Lin, Z., & Lv, H. P. Characterization of the key aroma compounds in Longjing tea using stir bar sorptive extraction (SBSE) combined with gas chromatography-mass spectrometry (GC–MS), gas chromatography-olfactometry (GC-O), odor activity value (OAV), and aroma recombination. Food Res Int,2020,130, 108908.https://doi.org/10.1016/j.foodres.2019.108908.

Comments 14: Line 150: 230°C; Line 150: 250°C

Response 14: Thanks for your careful check, we have modified the format of data and units accordingly.

Comments 15: Line 161: use presented instead of displayed

Response 15: Thanks to your suggestion, we have revised it in manuscript.

Comments 16: Line 161: use means instead of averages

Response 16: Thanks to your suggestion, we have revised it in manuscript.

Comments 17: Line 174: insert text space… 2035.25 μg/kg; Line 174: insert text space… 1799.02 μg/kg; Line 174: insert text space… 991.88 μg/kg; Line 175: 691.96 μg/kg; Line 176: 94.73 μg/kg; Line 176: 155.96 μg/kg; Line 178: 433.71 μg/kg; Line 179: 406.46 μg/kg; Line 182: 292.88 μg/kg

Response 17: Thanks for your careful check, we have modified the format of data and units accordingly.

Comments 18: Line 188-190: align ident text

Response 18: Have been modified

Comments 19: Line 190: > 100 μg/kg

Response 19: Thanks for your careful check, we have modified the format of data and units accordingly.

Comments 20: Line 194: In the last column of the table, start the first letter of each word with a capital letter, and insert text space where required.

Response 20: Have been modified

Comments 21: Line 317: insert a dot at the end of table title

Response 21: Have been modified

Comments 22: Line 324,325: align ident text; Line 355: align ident text;

Response 22: Have been modified

Comments 23: Line 370: [58–60]

Response 23: Have been modified

Comments 24: Line 356-357: align ident text

Response 24: Have been modified

Comments 25: Line 385: rewrite… Figure 6. The….

Response 25: Have been modified

Comments 26: Line 447: Journal names in references should be abbreviated (correct through this section)

Response 26: Thanks to the patient review of the reviewer, we have revised the reference format throughout.

Comments 27: Line 447: use italic text format for scientific names (correct through this section)

Response 27: Thanks to the patient review of the reviewers, we have revised format throughout.

Comments 28: Line 448: Only the volume of the journal should appear in italic text format; remove the issue number (correct through this section)

Response 28: Thanks to the patient review of the reviewer, we have revised the reference format throughout.

Comments 29: Line 449: References appear in which the authors' information is incomplete (correct through this section)

Response 29: Thanks to the patient review of the reviewer, we have completed the authors’ information.

Reviewer 2 Report

Comments and Suggestions for Authors

This manuscript is within the scope of the Journal. The authors studied the aroma volatile components of RZT green tea from a well identified growing region in China. The study was carried out by applying various techniques mainly based on GC-MS and other empirical-experimental methodologies. 

The methods and results were adequately described and discussed. The conclusions support the presented results. Adequate references have been provided. However, some minor issues have been raised that need very small corrections before acceptance.

1) line 300: here 37 compounds are mentioned to be reported in Table 2, while in Table 2 there are 36: am I making counting errors?, am I missing something?

2) Lines 300-304: some numerical inconsistencies are also present in this paragraph.

3) Table 2, row subheading – the IM h formalism should probably be corrected as IM f , and consequently the legend at the foot of the table needs to be adjusted;  .. identification method (IM) ….. mass spectrum (MS) …..

4) Figure 5.b – If the number “37” of compounds analyzed by GC-O … is ​​correct, then a species is missing in Table 2. If this were not the case, the opposite should apply.

5) lines 376 and 377 - Figure 6 shows SIX groups, not SEVEN: why?

The soundness of the paper is not questionable a priori, even in the presence of the inaccuracies noted.

Author Response

Comments 1: line 300: here 37 compounds are mentioned to be reported in Table 2, while in Table 2 there are 36: am I making counting errors?, am I missing something?

Response 1: Thank you for pointing this out. We agree with this comment. As a matter of fact, there should be 37 GC-O sniffing odorants in Table 2. unknown 2 was deleted due to the author's negligence, and it have been added. The changes can be seen in the last row of table 2 on page 15.

Comments 2: Lines 300-304: some numerical inconsistencies are also present in this paragraph.

Response 2: 

We really appreciate the reviewer's recommendation. Actually, the authors eliminated two compounds (unknown 1 and unknown 2) from the list of key odorants detected by GC-O because of their unknown chemical structure so as to ensure the reliability of the results. Considering the reviewer's advice, we believe these two unknowns ought to be on the list since their aroma characteristics and strength are quite evident.

Now we have classified these two unknowns according to their respective odor characteristics, and have modified this paragraph and Figure 4 accordingly. For details, please refer to Line 326-327 of the revised manuscript.

Comments 3: Table 2, row subheading – the IM h formalism should probably be corrected as IM f , and consequently the legend at the foot of the table needs to be adjusted;  .. identification method (IM) ….. mass spectrum (MS) …..

Response 3:Have been modified

Comments 4: Figure 5.b – If the number “37” of compounds analyzed by GC-O … is ​​correct, then a species is missing in Table 2. If this were not the case, the opposite should apply.

Response 4:

Thanks for the reviewer's suggestions. The authors have completed the missing structurally unknown compounds in Table 2. However, the two unknowns mentioned above could not be qualitatively and quantitatively examined due to the restricted experimental conditions.

The chemicals listed in Figure 5b were odorants that are shared by GC-O and OAV and have been clearly identified. Therefore, in order to ensure the accuracy of the study, we believe that the unknowns are not eligible for inclusion in Figure 5b. The author has revised the manuscript in line 368-374 as well as added the relevant remarks in Figure 5's legend to prevent needless misunderstandings.

 The revised contents are shown as follows: As depicted in Figure 5a, OAV and GC-O identified 25 and 37 key odorants, respectively, while the two groups mentioned above shared 17 common odorants. Therefore, the above 17 odorants including linalool, geranial, (Z)-jasmone, α-ionone, (E)-β-ionone, indole, linalool, geranial indole, hexanoic acid methyl ester, 2-pentylfuran, 3,5-octadien-2-one, β-cyclocitral, methyl salicylate, heptanal, hexanal, octanal, nonanal, 1-octen-3-ol, and 2-heptanone, verified by OAV and GC-O methods,  were considered as the key odorants of RZT.

Comments 5: lines 376 and 377 - Figure 6 shows SIX groups, not SEVEN: why?

Response 5:

Indeed, the reviewer's concern is really relevant. The 17 common odorants in Figure 6 were screened by OAV and GC-O. Based on their aroma characteristics, they can be divided into six categories: floral and sweet; fruity, citrus, and tropical; green and minty; aldehydic; cheesy; and earthy. In contrast to the seven aroma types in section 3.3, Figure 6 does not include a herbal type.

In fact, there were three herbal type odorants: 2-Ethyl-1-hexanol, (-)-Torreyol and α-Cadinol. However, the OAV of 2-Ethyl-1-hexanol was less than 1, while the OAV of (-)-Torreyol and α-Cadinol were not present because their odor thresholds were not available. Therefore, these 3 odorants had not been verified by the double screening of GC-O and OAV and were not shown in Figure 6.

Reviewer 3 Report

Comments and Suggestions for Authors

The manuscript investigates the volatile profiles and key odorants of Rizhao green tea (RZT), a renowned product from northern China, using advanced analytical techniques such as stir bar sorptive extraction coupled with gas chromatography-mass spectrometry (SBSE-GC-MS), odor activity value (OAV) analysis, and gas chromatography-olfactometry (GC-O). The study identifies 112 volatile compounds and highlights the key contributors to the tea's characteristic aroma, culminating in the creation of a molecular aroma wheel. This research is highly relevant and innovative, offering new insights into the aroma profile of RZT, which is distinct due to its unique cultivation environment. The use of integrated analytical and sensory methods significantly enhances the understanding of the aroma compounds contributing to the tea's quality. These findings are of value to the tea industry and researchers focused on flavor science. Comments and Suggestions for Improvement:

Abstract (Lines 14–30): Please include specific data in the abstract, such as the number of key odorants identified through dual OAV and GC-O analysis and examples of the most abundant volatiles. This will enhance clarity and reader engagement.

Introduction (Lines 31–97): Please elaborate on how the study builds upon previous research on green tea aroma, particularly highlighting the novelty of using SBSE-GC-MS and GC-O techniques for RZT.

Methods - Tea Samples (Lines 68–78): Please provide more detail on the criteria for selecting the six representative tea samples from the initial 31. Was there a sensory evaluation or compositional screening involved?

Methods - SBSE-GC-MS (Lines 93–117): Please clarify why specific parameters (e.g., 30-minute extraction time, 80°C temperature) were chosen for the SBSE process. Include references or preliminary optimization details.

Methods - GC-O Analysis (Lines 145–158): Please explain the panelist selection process for the GC-O analysis. Were they trained in tea aroma profiling?

Results - Volatile Compounds (Lines 165–191): Please discuss the potential biological pathways leading to the high abundance of compounds like geraniol and (Z)-3-hexenyl butanoate in RZT.

Results - OAV and GC-O Analysis (Lines 252–314): Please clarify the discrepancies between OAV and GC-O findings. Were any compounds with high OAV undetectable by GC-O, and why might this have occurred?

Results - Aroma Wheel (Lines 367–392): Please specify how the molecular aroma wheel could be applied in practice. For instance, could it assist in quality control or new product development in the tea industry?

Limitations and Future Directions: Please acknowledge the limitations of this study, such as the focus on a single growing region or potential variability due to different harvest years.

Please suggest directions for future research, such as exploring how seasonal changes impact aroma profiles or extending the analysis to other green tea varieties.

Author Response

Comments 1: Abstract (Lines 14–30): Please include specific data in the abstract, such as the number of key odorants identified through dual OAV and GC-O analysis and examples of the most abundant volatiles. This will enhance clarity and reader engagement.

Response 1: Thank you for pointing this out. We agree with this comment. Therefore, we have revised the abstract, and added some specific data into it. Details can be seen in line 22-27.

The contents added as :The key odorants identified in RZT were 17 volatiles, including linalool, geraniol, (Z)-jasmone, α-ionone, (E)-β-ionone, indole, hexanoic acid methyl ester, 2-pentylfuran, 3,5-octadien-2-one, β-cyclocitral, methyl salicylate, heptanal, hexanal, octanal, nonanal, 1-octen-3-ol, and 2-heptanone, as confirmed by OAV and GC-O. Among these 17 key odorants, linalool exhibited the highest aroma intensity (AI=3.67), while (E)-β-Ionone had the highest OAV (1625.22).

Comments 2: Introduction (Lines 31–97): Please elaborate on how the study builds upon previous research on green tea aroma, particularly highlighting the novelty of using SBSE-GC-MS and GC-O techniques for RZT.

Response 2: 

The authors have previously conducted a series of studies on the aroma quality of green tea, mainly including a review of the aroma components of China famous green tea; Optimization of SBSE-GC-MS method, research on key aroma components of fresh scent green tea, etc., and related research articles have also been published [1-4].

we had found that compared with SPME, SDE and SAFE, SBSE method has the advantages of large adsorption capacity, high sensitivity, good reproducibility and low detection limit.Additionally, the SBSE extraction procedure doesn't involve any organic solvents [4,5]. GC-O is a new generation of effective combination of human sensory smell analysis method, compared with traditional electronic nose, GC-O can evaluate The relevance of single compounds for the aroma assessed, as well as select aroma active compounds from complex mixtures[6]. Up to now,, we have successfully applied SBSE-GC-MS and GC-O analysis methods to the aroma quality analysis of green tea, yellow tea, white tea, dark tea etc. [3,7-9].

RZT is one of the representative famous teas in northern China, and its aroma analysis has important theoretical value for the study of tea quality in different regions of China. However, it is worth mentioning that the aroma analysis of RZT is still in a relatively basic stage, and there are no reports on the use of SBSE as well as GC-O method to analyze RZT. Therefore, this study is the first time to apply these two research methods to the research of RZT, which has certain innovation and research value.

[1] Wang, M., Zhu, Y., Zhang, Y., Shi, J., Lin, Z.,&Lv, H. A review of recent research on key aroma compounds in tea. Food Sci. 2019. 40(23),9. DOI:10.7506/spkx1002-6630-20181015-132.

[2] Wang, M., Zhu, Y., Zhang, Y., Shi, J., Lin, Z.,& Lv, H. Analysis of Volatile Compounds in “XihuLongjing” Tea by Stir Bar Sorptive Extraction Combine with Gas Chromatography-Mass Spectrometry. Food Sci. 2020,41(04):140-148.

[3] Wang, M. Q., Ma, W. J., Shi, J., Zhu, Y., Lin, Z., & Lv, H. P. Characterization of the key aroma compounds in Longjing tea using stir bar sorptive extraction (SBSE) combined with gas chromatography-mass spectrometry (GC–MS), gas chromatography-olfactometry (GC-O), odor activity value (OAV), and aroma recombination. Food Res Int. 2020,130, 108908.https://doi.org/10.1016/j.foodres.2019.108908.

[4] Wang, M., Study on Volatiles and Key Aroma Compounds of "Fresh Scent" Green Tea Based on SBSE-GC-MS. Chin.Aca. Agricul. Sci. 2020. 2-4.

[5] FAN W, SHEN H, XU Y., Quantification of volatile compounds in Chinese soy sauce aroma type liquor by stir bar sorptive extraction and gas chromatography-mass spectrometry[J]. J. Sci. Agric. 2011.91(7): 1187-1198. DOI:10.1002/jsfa.4294.

[6] Wardencki, W., Chmiel, T., & Dymerski, T. Gas chromatography-olfactometry (GC-O), electronic noses (e-noses) and electronic tongues (e-tongues) for in vivo food flavour measurement. Inst. Assess. Food Sens. Qqual. 2013,195-229.DOI:10.1533/9780857098856.2.195.

[7] Shi, Y., Wang, M., Dong, Z., Zhu, Y., & Lv, H. Volatile components and key odorants of chinese yellow tea (camellia sinensis). LWT- Food Sci. Technol. 2021. 146(33), 111512.

[8] Yan, H., Li, W. X., Zhu, Y. L., Lin, Z. Y., Chen, D., Zhang, Y., Lv, H., Dai, W., Ni, D., Lin, Z., & Zhu, Y. Comprehensive comparison of aroma profiles and chiral free and glycosidically bound volatiles in Fujian and Yunnan white teas. Food Chem, 2024.448, 139067.

[9] Ma, W., Zhu, Y., Ma, S., Shi, J., Yan, H., Lin, Z., & Lv, H. Aroma characterisation of Liu-pao tea based on volatile fingerprint and aroma wheel using SBSE-GC–MS. Food Chem. 2023, 414, 135739.DOI:10.1016/j.foodchem.2023.135739.

Comments 3: Methods - Tea Samples (Lines 68–78): Please provide more detail on the criteria for selecting the six representative tea samples from the initial 31. Was there a sensory evaluation or compositional screening involved?

Response 3: 

Thanks to the reviewer's suggestion, we have added the reference standards and process details of sensory evaluation to the revised draft (line 77-84), and the added details are listed below:

sensory evaluation of the above 31 RZT samples was performed according to the ”Tea Sensory Evaluation Method” (GB/T 23776–2018) and “Tea Sensory Evaluation Terminology” (GB/T 14487–2017). The sensory evaluation was conducted by 5 tea experts who are qualified as senior tea assessors, all of whom have more than 10 years of experience in tea sensory evaluation. Experts rated the tea samples for appearance, soup color, aroma, taste and the bottom of the leaves, and those with sensory scores < 89 were removed. Ultimately, six tea samples that exhibited the most representative and quintessential flavor of RZT were finally selected as the subjects of this study.

Comments 4: Methods - SBSE-GC-MS (Lines 93–117): Please clarify why specific parameters (e.g., 30-minute extraction time, 80°C temperature) were chosen for the SBSE process. Include references or preliminary optimization details.

Response 4: 

The author had previously used single factor experiment combined with orthogonal experiment to optimize parameters of SBSE method, such as stirring rod type, extraction temperature, time, tea-water ratio, rotational speed and ionic strength.The result showed that the volatile aroma compounds in“Xihu Longjing” tea can be extracted and adsorbed to the greatest extent under the condition of PDMS twister, a 60:1 tea-water ratio with 5% ionic strength, 80℃hot water, an extraction time of 90 minutes at an agitation rate of 1250rpm. The overall accuracy of the experiment (the high concentration recovery was 115.11%, the low concentration recovery was 108.99%) and repeatability (RSD=8.23%) were satisfactory. This method has been successfully applied to the aroma quality analysis of “Xihu Longjing tea”[2].

Later, Zhu Yin, one of the authors of this paper, found that the extraction time of 90min would cause excessive water content in the PDMS twister, resulting in occasional failure of the TDU program and column loss during the detection process. Therefore, the authors further optimized the method to 80℃ and 30min extraction time. This method has been successfully applied to the aroma analysis of various tea products[3–5].

  • Wang, M., Zhu, Y., Zhang, Y., Shi, J,m Lin, Z., & Lv, H. analysis of volatile compounds in “xihulongjing”tea by stir bar sorptive extraction combine with gas chromatography-mass spectrometry. Food Sci. 2020.41(4), 9. DOI: CNKI:SUN:SPKX.0.2020-04-019
  • Wang, M., Ma, W., Shi, J., Zhu, Y., Lin, Z., & Lv, H.Characterization of the key aroma compounds in Longjing tea using stir bar sorptive extraction (SBSE) combined with gas chromatography-mass spectrometry (GC–MS), gas chromatography-olfactometry (GC-O), odor activity value (OAV), and aroma recombination. Food Res Int. 2020,130, 108908.https://doi.org/10.1016/j.foodres.2019.108908.
  • You Q, Shi Y, Zhu Y , Yang G, Yan H.,Lin, Z., & Lv H. Effect of Different Processing Technologies on the Key Aroma-Active Compounds of Green Tea. Food Sci. 2023, 44(8), 7. (in Chinese). DOI: 10.7506/spkx1002-6630-20220718-204.
  • Ma, W., Zhu, Y., Ma, S., Shi, J., Yan, H., Lin, Z., & Lv, H. Aroma characterisation of Liu-pao tea based on volatile fingerprint and aroma wheel using SBSE-GC–MS. Food Chem. 2023, 414, 135739.DOI:10.1016/j.foodchem.2023.135739.

You, Q., Liu, J., He, W., Zhu, Y., Lin, Z., & Lv, H. Aroma Components of Huiming Tea. Food Chem. 2023, 44(24),253.DOI:10.7506/spkx1002-6630-20230330-311.

Comments 5: Methods - GC-O Analysis (Lines 145–158): Please explain the panelist selection process for the GC-O analysis. Were they trained in tea aroma profiling?

Response 5:

The assessors were tea-tasters or tea experts who authenticated by professional organizations and have worked in the field of sensory evaluation for at least four years. After the establishment of the group, all panelists have been trained in the sniffing of odor standards and different types of tea for more than 90 hours, the training process is detailed in the references [1]:

Firstly, each assessor was well-trained for more than 90 h in order to get familiar with the different odor descriptions and odor strengths by using a series of standard solutions with different concentrations. The applied standards were listed as follows, geraniol and linalool represented “floral” odor, β-damascenone represented “sweet and honey-like” odor, hexanal represented “green and fresh” odor, 2,3,5-trimethylpyrazine represented “roast and baked” odor, (Z)-hex-3-en-1-yl acetate represented “fruity” odor, 2-acetyl pyrazine represented “hazelnut and coffee” odor, menthol represented “minty” odor, and linalool oxide (pyranoid) represented “earthy” odor, and the above standards were diluted to at least 5 concentration gradients with absolute ethyl alcohol solution. Next, the assessors were familiar with the GC-O technique and attempted the GC-O analysis of different types of tea samples (green, black and dark teas) for 30 h during the last one month before the true GC-O analysis.

 [1] Zhu, Y., Lv, H. P., Shao, C. Y., Kang, S., Zhang, Y., Guo, L., Dai, W., Tan, J., Peng, Q., & Lin, Z. Identification of key odorants responsible for chestnut-like aroma quality of green teas. Food Res Int. 2018. 108, 74-82.https://doi.org/10.1016/j.foodres.2018.03.026.

Comments 6: Results - Volatile Compounds (Lines 165–191): Please discuss the potential biological pathways leading to the high abundance of compounds like geraniol and (Z)-3-hexenyl butanoate in RZT.

Response 6:

In response to the reviewer's recommendations, we have included the potential causes and mechanisms for the production of these two highly volatiles in the updated manuscript. Specifics can be found on page 5, lines 196–206.Furthermore, Figure 3 and lines 238-262 of the manuscript also hypothesize on the causes of the two volatiles' high abundance.

The additions are as follows:

 It was discovered that (Z)-3-hexanoic acid hexenyl ester could increase the expression of the essential enzyme genes in the synthesis pathway, CsADH1, CsADH3, and CsLOX3, to strengthen the tea plant's tolerance to cold [15].

 According to earlier research, cold stress can also cause a rise in the amounts of geraniol and linalool [20]. Although the exact process by which geraniol and linalool are inducted in tea is still unknown, it is hypothesized that the rise in these volatiles is connected to the alteration of the tea tree's metabolic pathway at low temperatures.

[15] Zhang, X., Li, J., Chen, X., Wang, W., Li, F., & Ma, Y. Effects of leaf acetate on physiology and biochemistry of cold tolerance of tea plant under low temperature stress. Jiangsu Agricul Sci, 2021.49(24), 6.DOI:10.15889/j.issn.1002-1302.2021.24.022

[20] Zhao, M., Wang, L., Wang, J., Zhang, J., Zhang, N., Lei, L., Gao, T., Jing, T., Zhang, S., Wu, B., Hu, Y., Wan, X., Schwab, W., & Song, C. Induction of priming by cold stress via inducible volatile cues in neighboring tea plants. J.Int.Plant Biolo. 2020. 62(10), 1461-1468.https://doi.org/10.1111/jipb.12937.

Comments 7: Results - OAV and GC-O Analysis (Lines 252–314): Please clarify the discrepancies between OAV and GC-O findings. Were any compounds with high OAV undetectable by GC-O, and why might this have occurred?

Response 7:

The difference analysis of the OAV and GC-O results can be seen in Chapter 3.4. Just as the reviewers said, there were indeed compounds with high OAV deficiency that cannot be detected by GC-O, such as decanal and phytol. We speculate that there were two main reasons for this phenomenon:

1: There are interaction effects between volatiles (section 3.4 also mentioned this). The volatiles in tea interact with each other rather than being alone. Some volatiles with OAV less than 1 are perceived by people because of the synergistic effect with other volatiles. However, some volatiles with OAV greater than 1 may be unaware of their aroma properties due to the antagonism between them and other volatiles[1].

2: Furthermore, the thresholds for certain volatile components may vary depending on the media. The threshold in air determines a volatile component's odor in GC-O analysis, whereas the threshold in water determines the OAV value[2].

In conclusion, the combination of OAV and GC-O can assess the quality of tea aroma more thoroughly.

[1] Zhu, J. C., Chen, F., Wang, L., Niu, Y., & Xiao, Z. Evaluation of the synergism among volatile compounds in oolong tea infusion by odour threshold with sensory analysis and e-nose. Food Chem. 2016. 221, 1484-1490.

[2] Gong, X., Han, Y., Zhu, J., Hong, L., Zhu, D., Liu, J., ... & Xiao, Z. Identification of the aroma-active compounds in Longjing tea characterized by odor activity value, gas chromatography-olfactometry, and aroma recombination. Int. J. Food. Proper,2017. 20(sup1), S1107-S1121.

Comments 8: Results - Aroma Wheel (Lines 367–392): Please specify how the molecular aroma wheel could be applied in practice. For instance, could it assist in quality control or new product development in the tea industry?

Response 8:

Thanks to the reviewers for their effective suggestions, we have given examples of the application of molecular aroma wheels in production practice, as shown in the details below. In addition, we have added corresponding content in Chapter 3.5 of the revised draft.

The molecular aroma wheel is a valuable tool that can be applied in various practical ways within the tea industry, particularly in quality control and new product development.

The Molecular Aroma Wheel can provide a detailed and standardized framework for evaluating tea aroma by integrating sensory descriptions and chemical compositions. For example, with reference to the six aromas and 17 key odorants of RZT's aroma wheel, tea processors can use tools to check whether the aroma of the product is consistent with the aroma wheel, thus ensuring product consistency and representation.

In new product development, the molecular aroma wheel can guide the development of tea products with specific aroma characteristics to meet consumer preferences. For example, for the customer's different preferences for tea flavor: highlight floral or fragrant. Tea processors can use the aroma types in the aroma wheel to find the corresponding key odorants, and improve the content of a key odorant by adjusting the processing conditions, so as to obtain tea products with outstanding aroma types.

In conclusion, the aroma wheel is a scientific and visual method of aroma characteristics analysis, which has important reference value for improving the quality and consistency of tea products.

Comments 9: Limitations and Future Directions: Please acknowledge the limitations of this study, such as the focus on a single growing region or potential variability due to different harvest years.

Response 9:

We appreciate the reviewer's suggestion regarding this issue. In fact, there are indeed some limitations in our experiment, which have been elaborated below. Additionally, we have incorporated the limitations of this experiment into the original text, which can be found in line 433-443.

While this study has yielded significant findings, several limitations must be acknowledged. Firstly, commercially available tea samples were used in this study, which may affect result reliability due to the slight differences of the production process. Additionally, the precursors and the biochemical pathways leading to the formation of these key odorants remain obscure, as do the regulatory mechanisms under the specific conditions of high latitude and low temperature. To address these limitations, and conduct a more comprehensive investigation into RZT’s key odorants, future research will adopt a refined sampling strategy by processing the collected fresh tea leaves in a standardized technology. Moreover, The sophisticated analysis technologies, including metabonomics and transcriptomics, will be essential to provide deeper insights into the underlying mechanisms

Comments 10: Please suggest directions for future research, such as exploring how seasonal changes impact aroma profiles or extending the analysis to other green tea varieties.

Response 10:

Thanks to the reviewers for their valuable suggestions on future research, we have added the idea of future research to the revised draft (chapter 4: conclusions. Line 457), the details are as follows:

Future research will examine how the amount of aroma precursors in tea leaves varies from winter to spring in order to provide light on how temperature and seasonal variations affect RZT's aroma qualities in high latitudes. In addition, we will broaden the tea varieties and sample areas, and strive to provide a more thorough theoretical foundation for the investigation of the aroma characteristics of green tea in northern China.

Round 2

Reviewer 3 Report

Comments and Suggestions for Authors

Dear Authors,

This study makes a significant contribution to tea science and food chemistry. It provides a solid foundation for understanding the sensory attributes of RZT and serves as a model for similar analyses of other tea varieties. However, as noted during the review process, some limitations were identified and addressed in the article. Another improvement was the inclusion of a more robust scientific discussion. Additionally, the incorporation of updated references further supports the findings presented in the study.

Given the revisions made, I consider the article suitable for publication.